

# Value of lymph node ratio as a prognostic factor of recurrence in medullary thyroid cancer

Weijing Hao[1,*], Jingzhu Zhao[1,*], Fengli Guo[1,2], Pengfei Gu[1], Jinming Zhang[1], Dongmei Huang[1], Xianhui Ruan[1], Yu Zeng[1], Xiangqian Zheng[1] and Ming Gao[1,3,4]

[1] Department of Thyroid and Neck Tumor, Tianjin Medical University Cancer Institute and Hospital, National Clinical Research Center for Cancer, Key Laboratory of Cancer Prevention and Therapy, Tianjin's Clinical Research Center for Cancer, Tianjin, China
[2] Department of Thyroid and Breast Surgery, Binzhou Medical University Hospital, Binzhou, Shandong, China
[3] Department of Thyroid and Breast Surgery, Tianjin Union Medical Center, Tianjin, China
[4] Tianjin Key Laboratory of General Surgery in Construction, Tianjin Union Medical Center, Tianjin, China
* These authors contributed equally to this work.

Corresponding authors
Xiangqian Zheng,
xzheng05@tmu.edu.cn
Ming Gao,
headandneck2008@126.com

## ABSTRACT

**Background and Objectives:** The purpose of this study is to evaluate the relationship between lymph node status (the number of resected lymph nodes; the number of metastatic lymph nodes, LNM, and lymph node ratio, LNR) and biochemical recurrence, disease-free survival (DFS), as well as overall survival (OS) in medullary thyroid carcinoma (MTC).
**Methods:** This study enrolled MTC patients at Tianjin Medical University Cancer Institute and Hospital between 2011 and 2019. We used Logistic regression analysis, Cox regression models and Kaplan-Meier test to identify risk factors influencing biochemical recurrence, DFS, and OS.
**Results:** We identified 160 patients who satisfied the inclusion criteria from 2011 to 2019. We used ROC analysis to define the cut-off value of LNR with 0.24. Multifocality, preoperative calcitonin levels, pathologic N stage, resected lymph nodes, LNM, LNR, and the American Joint Committee on Cancer (AJCC) clinical stage were significant ($P < 0.05$) prognostic factors influencing biochemical cure. In univariable analyses, gross extrathyroidal extension, preoperative calcitonin levels, pathologic T classification, pathologic N stage, resected lymph nodes, LNM, LNR, AJCC clinical stage, and biochemical cure were significant ($P < 0.05$) factors of DFS. When the multivariable analysis was performed, LNR was identified as predictor of DFS (HR = 4.818, 95% CI [1.270–18.276]). Univariable Cox regression models reflected that tumor size, pathologic N stage, and LNR were predictor of OS. Furthermore, multivariable analysis manifested that LNR was predictor of OS (HR = 10.061, 95% CI [1.222–82.841]).
**Conclusions:** This study illustrated that LNR was independent prognostic factor of DFS and OS in MTC. In addition, LNR influenced biochemical cure. Further investigations are needed to determine the optimal cut-off value for predicting prognosis.

## INTRODUCTION

Medullary thyroid carcinoma (MTC) is a rare C-cell-derived neuroendocrine malignancy. It accounts for 1–2% of all thyroid cancers in the United States. Most MTC cases (75%) are sporadic, while 25% are familial and associated with germ-line mutations (*Erovic et al., 2012*). MTC cells do not concentrate radioactive iodine and thyroid stimulating hormone insensitivity (*Jin & Moley, 2016*). Thus, surgical treatment is the mainstay of therapy. In MTC, lymph nodes metastasis can be detected in up to 75% of the patients (*Moses et al., 2021*; *Polistena et al., 2017*). In addition, 10-year survival rates for MTC are below 80%, and the cervical metastasis had significant influence on survival for MTC (*Bhattacharyya, 2003*; *Brierley et al., 1996*). The prognosis of MTC varied and disease related factors included age, gender, lymph node metastases, calcitonin levers, distant metastases, and response to initial treatment (*Wells et al., 2015*).

The current American Joint Committee on Cancer (AJCC) staging system for MTC categorizes lymph nodes (LN) status as N0 (no positive nodes), N1a (positive nodes in the central neck compartment), and N1b (positive nodes in the lateral neck) (*Amin et al., 2017*). It does not take into account the number of resected and positive nodes. Typically, the treatment of MTC involves routine central compartment dissection, and lateral neck dissection is recommended for patients with structural evidence of lateral compartment metastasis or with high preoperative calcitonin levels. Almost all patients will undergo some kind of lymph node dissection. Whether the lymph nodes status has any predictive value is not well illustrated.

The purpose of this study is to evaluate the relationship between lymph node status (the number of resected lymph nodes; the number of metastatic lymph nodes, LNM, and lymph node ratio, LNR) and biochemical recurrence, disease-free survival (DFS), as well as overall survival (OS). In addition, we test to investigate the optimal LNR cut-off value that best predicts the outcome.

## MATERIALS AND METHODS

We retrospectively searched the databases for patients with MTC at Tianjin Medical University Cancer Institute and Hospital between 2011 and 2019. The present study was approved by the Institutional Review Board (bc2022191). Written informed consent was obtained from the patients.

All patients undergoing primary surgical treatment for MTC were included. Patients were excluded if they had pathologically positive resection margin, distant metastasis, or a history of thyroidectomy. Additionally, patients with a family history of MTC, a history of other malignancy, or incomplete data were not included.

Patient demographics, clinicopathologic factors, and survival outcomes were recorded. All the patients were operated on total thyroidectomy or hemithyroidectomy with central

or both central and lateral compartment dissection considering preoperative imaging and calcitonin levels. Serum calcitonin was measured using the immunoradiometric assay. All the specimens in this study were analyzed by two or more dedicated head and neck pathologists. Recurrence was defined as the appearance of disease with pathology-confirmed local or distant disease detected by imaging scans 3 months after surgery. A biochemical cure was described as an abnormal preoperative calcitonin level declining within the reference range within 6 months after surgery. Patients' follow-up primarily included neck ultrasound/CT and calcitonin levels.

We evaluated (1) the number of resected lymph nodes: 0 to 10, and greater than ten nodes; (2) LNM and (3) LNR, the number of metastatic lymph nodes divided by the number of resected lymph nodes. The nodal status was investigated in terms of its association with all the mentioned demographic, pathological, and prognostic variables. We used ROC analysis to define the cut-off value of LNR that best reflected prognosis.

Statistical analysis was performed using SPSS software (version 20.0; IBM, Chicago, IL, USA). Chi-squared analysis was used to compare frequencies between groups. Logistic regression analysis was used to identify risk factors influencing biochemical recurrence. Univariable and multivariable Cox regression models were applied to find risk factors influencing structural recurrence. Survival analysis was performed using Kaplan-Meier test. $P < 0.05$ was considered to indicate statistically significant differences.

## RESULTS

### Baseline characteristics of the study population

We identified 160 patients who satisfied the inclusion criteria from 2011 to 2019. Demographic data are displayed in Table 1. The median age at the time of diagnosis was 52 years (14–73), and the majority of patients were female (90, 56.3%). The mean size of the largest tumor diameter was 1.79 cm, and 61 (38.1%) patients had an extrathyroidal extension. A total of 13 (8.1%) patients had bilateral tumors and 44 (27.5%) patients had multifocal tumors. Only central LN dissection was conducted in 77 (48.1%) patients. Meanwhile, central and lateral LN dissection was conducted in 82 (51.3%) patients. Approximately half of the patients had advanced stage MTC (stages III-IV, 59.4%). Positive lymph nodes were identified in 89 (55.6%) of cases. The median length of follow-up was 51 months (10–114 months). Structural recurrence was identified in 24 (15.0%) patients, and 12 (7.5%) patients died at the end of the study period. DFS and OS for the entire cohort were 83.1% and 91.3% at 5 years, respectively.

We used ROC analysis to define the cut-off value of LNR, and 0.24 was determined as the cut-off level with the highest predictive performance. The cumulative survivals of the cohort are shown in Fig. 1.

### Association of resected lymph nodes, LNM, LNR, and pathologic N classification with patient and tumor characteristics

The clinicopathologic characteristics of resected lymph nodes group, LNM group, LNR group, and pathologic N classification are shown in Tables 2 and 3. There were no

**Table 1 Clinicopathologic characteristics of 160 medullary thyroid carcinoma patients.**

| Features | N | Percentage |
|---|---|---|
| Total | 160 | 100 (%) |
| Age: years, median ± SD | 52 ± 12.0 | |
| Gender | | |
| Male | 70 | 43.8 |
| Female | 90 | 56.3 |
| Tumor size (cm) | | |
| ≤2 cm | 115 | 71.9 |
| >2 cm | 45 | 28.1 |
| Multifocality | | |
| Yes | 44 | 27.5 |
| No | 116 | 72.5 |
| Extrathyroidal extension | | |
| Yes | 61 | 38.1 |
| No | 99 | 61.9 |
| Bilateral | | |
| Yes | 13 | 8.1 |
| No | 147 | 91.9 |
| Pathologic T classification | | |
| pT1 | 74 | 46.3 |
| pT2 | 21 | 13.1 |
| pT3 | 39 | 24.4 |
| pT4 | 26 | 16.3 |
| Pathologic N classification | | |
| pN0 | 71 | 44.4 |
| pN1a | 22 | 13.8 |
| pN1b | 67 | 41.9 |
| Resected lymph nodes: median (IQR) | 14 (3–32) | |
| Metastasized lymph nodes: median (IQR) | 1 (0–7) | |
| Lymph node ratio: median (IQR) | 0.123 (0–0.377) | |
| AJCC clinical stage | | |
| I | 40 | 25.0 |
| II | 25 | 15.6 |
| III | 18 | 11.3 |
| IV | 77 | 48.1 |
| Preoperative calcitonin: median (IQR) | 521 (129–1,555) | |
| Lymph node dissection | | |
| Only central LND | 77 | 48.1 |
| Central and lateral LND | 82 | 51.3 |
| Not done | 1 | 0.6 |
| Biochemical cure | | |
| Yes | 84 | 60.9 |

| Table 1 (continued) | | |
| --- | --- | --- |
| Features | N | Percentage |
| No | 54 | 39.1 |
| Unknown | 22 | 13.8 |
| Recurrence | | |
| Yes | 24 | 15.0 |
| No | 135 | 84.4 |
| Unknown | 1 | 0.6 |
| Death | | |
| Yes | 12 | 7.5 |
| No | 147 | 91.9 |
| Unknown | 1 | 0.6 |
| Follow-up duration: months, median (IQR) | 51 (36–72) | |

**Note:**
SD, standard deviation; IQR, interquartile range.

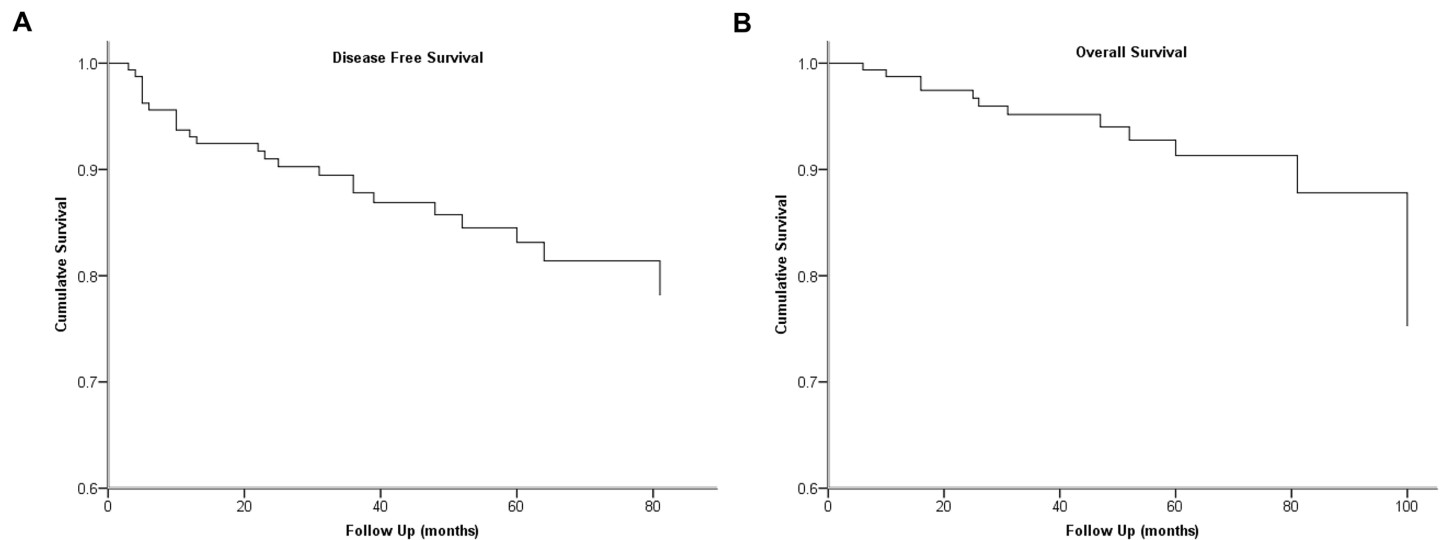

**Figure 1** Kaplan-Meier survival plots present cumulative disease-free survival (A) and overall survival (B) of the cohort.

significant differences in age, capsule invasion, and bilateral between groups, while prognostic factors varied.

## Prognostic factors influencing biochemical cure

In chi-squared analysis, multifocality, preoperative calcitonin levels, pathologic N stage, resected lymph nodes, LNM, LNR, and AJCC clinical stage were significant ($P < 0.05$) prognostic factors influencing biochemical cure (Table 4). While logistic regression analysis did not identify independent risk factors (Table 5).

**Table 2 Clinicopathologic characteristics according to resected lymph nodes group and metastasized lymph nodes group.**

| Variables | Resected lymph nodes | | | Metastasized lymph nodes | | |
|---|---|---|---|---|---|---|
| | ≤10 | >10 | P value | ≤10 | >10 | P value |
| Age (years) | | | 0.118 | | | 0.636 |
| ≤50 | 30 (40.0) | 44 (52.4) | | 63 (47.4) | 11 (42.3) | |
| >50 | 45 (60.0) | 40 (47.6) | | 70 (52.6) | 15 (57.7) | |
| Gender | | | **0.036** | | | **0.013** |
| Male | 26 (34.7) | 43 (51.2) | | 52 (39.1) | 17 (65.4) | |
| Female | 49 (65.3) | 41 (48.8) | | 81 (60.9) | 9 (34.6) | |
| Tumor size (cm) | | | **0.004** | | | **0.027** |
| ≤2 cm | 62 (82.7) | 52 (61.9) | | 100 (75.2) | 14 (53.8) | |
| >2 cm | 13 (17.3) | 32 (38.1) | | 33 (24.8) | 12 (46.2) | |
| Multifocality | | | **0.016** | | | **0.021** |
| Yes | 14 (18.7) | 30 (35.7) | | 32 (24.1) | 12 (46.2) | |
| No | 61 (81.3) | 54 (64.3) | | 101 (75.9) | 14 (53.8) | |
| Extrathyroidal extension | | | 0.059 | | | 0.076 |
| Yes | 23 (30.7) | 38 (45.2) | | 47 (35.3) | 14 (53.8) | |
| No | 52 (69.3) | 46 (54.8) | | 86 (64.7) | 12 (46.2) | |
| Bilateral | | | 0.512 | | | 0.063 |
| Yes | 5 (6.7) | 8 (9.5) | | 8 (6.0) | 5 (19.2) | |
| No | 70 (93.3) | 76 (90.5) | | 125 (94.0) | 21 (80.8) | |
| Preoperative calcitonin | | | **<0.001** | | | **<0.001** |
| ≤300 ng/L | 46 (63.0) | 13 (16.2) | | 59 (46.1) | 0 (0.0) | |
| >300 ng/L | 27 (37.0) | 67 (83.8) | | 69 (53.9) | 25 (100.0) | |
| Pathologic T classification | | | **0.031** | | | 0.057 |
| T1/T2 | 51 (68.0) | 43 (51.2) | | 83 (62.4) | 11 (42.3) | |
| T3/T4 | 24 (32.0) | 41 (48.8) | | 50 (37.6) | 15 (57.7) | |
| Pathologic N classification | | | **<0.001** | | | **<0.001** |
| pN0 | 58 (77.3) | 12 (14.3) | | 70 (52.6) | 0 (0.0) | |
| pN1a | 15(20.0) | 7 (8.3) | | 21 (15.8) | 1 (3.8) | |
| pN1b | 2 (2.7) | 65 (77.4) | | 42 (31.6) | 25 (96.2) | |
| AJCC clinical stage | | | **<0.001** | | | **<0.001** |
| I/II | 52 (69.3) | 12 (14.3) | | 64 (48.1) | 0 (0.0) | |
| III/IV | 23 (30.7) | 72 (85.7) | | 69 (51.9) | 26 (100.0) | |
| Recurrence | | | **<0.001** | | | **0.025** |
| Yes | 3 (4.0) | 21 (25.3) | | 16 (12.0) | 8 (32.0) | |
| No | 72 (96.0) | 62 (74.7) | | 117 (88.0) | 17 (68.0) | |
| Death | | | **0.001** | | | 0.217 |
| Yes | 0 (0.0) | 12 (14.5) | | 8 (6.1) | 4 (15.4) | |
| No | 75 (100.0) | 71 (85.5) | | 124 (93.9) | 22 (84.6) | |

**Note:**
The bold part of the P value represents P < 0.05.

**Table 3 Clinicopathologic characteristics according to lymph node ratio group and pathologic N stage group.**

| Variables | LNR | | | Pathologic N stage group | | | |
|---|---|---|---|---|---|---|---|
| | ≤0.24 | >0.24 | *P* value | pN0 | pN1a | pN1b | *P* value |
| Age (years) | | | 0.577 | | | | 0.501 |
| ≤50 | 42 (50.6) | 28 (45.9) | | 33 (46.5) | 8 (36.4) | 34 (50.7) | |
| >50 | 41 (49.4) | 33 (54.1) | | 38 (53.5) | 14 (63.6) | 33 (49.3) | |
| Gender | | | **0.015** | | | | **0.005** |
| Male | 28 (33.7) | 33 (54.1) | | 21 (29.6) | 11 (50.0) | 38 (56.7) | |
| Female | 55 (66.3) | 28 (45.9) | | 50 (70.4) | 11 (50.0) | 29 (43.3) | |
| Tumor size (cm) | | | 0.733 | | | | 0.163 |
| ≤2 cm | 58 (69.9) | 41 (67.2) | | 52 (73.2) | 19 (86.4) | 44 (65.7) | |
| >2 cm | 25 (30.1) | 20 (32.8) | | 19 (26.8) | 3 (13.6) | 23 (34.3) | |
| Multifocality | | | **<0.001** | | | | **0.003** |
| Yes | 13 (15.7) | 26 (42.6) | | 12 (16.9) | 4 (18.2) | 28 (41.8) | |
| No | 70 (84.3) | 35 (57.4) | | 59 (83.1) | 18 (81.8) | 39 (58.2) | |
| Extrathyroidal extension | | | 0.094 | | | | **0.012** |
| Yes | 28 (33.7) | 29 (47.5) | | 18 (25.4) | 11 (50.0) | 32 (47.8) | |
| No | 55 (66.3) | 32 (52.5) | | 53 (74.6) | 11 (50.0) | 35 (52.2) | |
| Bilateral | | | 0.242 | | | | 0.604 |
| Yes | 5 (6.0) | 7 (11.5) | | 5 (7.0) | 1 (4.5) | 7 (10.4) | |
| No | 78 (94.0) | 54 (88.5) | | 66 (93.0) | 21 (95.5) | 60 (89.6) | |
| Preoperative calcitonin | | | **0.005** | | | | **<0.001** |
| ≤300 ng/L | 38 (46.9) | 14 (23.7) | | 37 (53.6) | 15 (71.4) | 8 (12.5) | |
| >300 ng/L | 43 (53.1) | 45 (76.3) | | 32 (46.4) | 6 (28.6) | 56 (87.5) | |
| Pathologic T classification | | | 0.078 | | | | **0.006** |
| T1/T2 | 53 (63.9) | 30 (49.2) | | 52 (73.2) | 10 (45.5) | 33 (49.3) | |
| T3/T4 | 30 (36.1) | 31 (50.8) | | 19 (26.8) | 12 (54.5) | 34 (50.7) | |
| Pathologic N classification | | | **<0.001** | | | | |
| pN0 | 55 (66.3) | 0 (0.0) | | | | | |
| pN1a | 5 (6.0) | 17 (27.9) | | | | | |
| pN1b | 23 (27.7) | 44 (72.1) | | | | | |
| AJCC clinical stage | | | **<0.001** | | | | **<0.001** |
| I/II | 51 (61.4) | 0 (0.0) | | 65 (91.5) | 0 (0.0) | 0 (0.0) | |
| III/IV | 32 (38.6) | 61 (100.0) | | 6 (8.5) | 22 (100.0) | 67 (100.0) | |
| Recurrence | | | **<0.001** | | | | **<0.001** |
| Yes | 4 (4.8) | 19 (31.7) | | 2 (2.8) | 3 (13.6) | 19 (28.8) | |
| No | 79 (95.2) | 41 (68.3) | | 69 (97.2) | 19 (86.4) | 47 (71.2) | |
| Death | | | **<0.001** | | | | **<0.001** |
| Yes | 1 (1.2) | 11 (18.0) | | 0 (0.0) | 1 (4.5) | 11 (16.4) | |
| No | 81 (98.8) | 50 (82.0) | | 70 (100.0) | 21 (95.5) | 56 (83.6) | |

Notes:
LNR, lymph node ratio. The bold part of the *P* value represents *P* < 0.05.

**Table 4 Clinicopathologic characteristics according to biochemical cure.**

| Variables | Biochemical cure | | | P value |
|---|---|---|---|---|
| | **Total** | **Yes** | **No** | |
| Age (years) | | | | 0.237 |
| ≤50 | 68 (49.3) | 38 (45.2) | 30 (55.6) | |
| >50 | 70 (50.7) | 46 (54.8) | 24 (44.4) | |
| Gender | | | | 0.215 |
| Male | 60 (43.5) | 33 (39.3) | 27 (50.0) | |
| Female | 78 (56.5) | 51 (60.7) | 27 (50.0) | |
| Tumor size (cm) | | | | 0.198 |
| ≤2 cm | 98 (71.0) | 63 (75.0) | 35 (64.8) | |
| >2 cm | 40 (29.0) | 21 (25.0) | 19 (35.2) | |
| Multifocality | | | | **0.030** |
| Yes | 37 (26.8) | 17 (20.2) | 20 (37.0) | |
| No | 101 (73.2) | 67 (79.8) | 34 (63.0) | |
| Extrathyroidal extension | | | | 0.242 |
| Yes | 53 (38.4) | 29 (34.5) | 24 (44.4) | |
| No | 85 (61.6) | 55 (65.5) | 30 (55.6) | |
| Bilateral | | | | 0.082 |
| Yes | 10 (7.2) | 3 (3.6) | 7 (13.0) | |
| No | 128 (92.8) | 81 (96.4) | 47 (87.0) | |
| Pathologic T classification | | | | 0.096 |
| pT1/T2 | 81 (58.7) | 54 (64.3) | 27 (50.0) | |
| pT3/T4 | 57 (41.3) | 30 (35.7) | 27 (50.0) | |
| Preoperative calcitonin | | | | **0.001** |
| ≤300 ng/L | 51 (37.0) | 40 (47.6) | 11 (20.4) | |
| >300 ng/L | 87 (63.0) | 44 (52.4) | 43 (79.6) | |
| Pathologic N classification | | | | **<0.001** |
| pN0 | 60 (43.5) | 51 (60.7) | 9 (16.7) | |
| pN1a | 21 (15.2) | 12 (14.3) | 9 (16.7) | |
| pN1b | 57 (41.3) | 21 (25.0) | 36 (66.7) | |
| Resected lymph nodes | | | | **<0.001** |
| ≤10 | 65 (47.1) | 51 (60.7) | 14 (25.9) | |
| >10 | 73 (52.9) | 33 (39.3) | 40 (74.1) | |
| Metastasized lymph nodes | | | | **<0.001** |
| ≤10 | 118 (85.5) | 80 (95.2) | 38 (70.4) | |
| >10 | 20 (14.5) | 4 (4.8) | 16 (29.6) | |
| Lymph node ratio | | | | **<0.001** |
| ≤0.24 | 73 (57.9) | 58 (76.3) | 15 (30.0) | |
| >0.24 | 53 (42.1) | 18 (23.7) | 35 (70.0) | |
| AJCC clinical stage | | | | **<0.001** |
| I/II | 54 (39.1) | 48 (57.1) | 6 (11.1) | |
| III/IV | 84 (60.9) | 36 (42.9) | 48 (88.9) | |

| Variables | Biochemical cure | | | P value |
|---|---|---|---|---|
| | Total | Yes | No | |
| Recurrence | | | | **0.002** |
| Yes | 20 (14.5) | 6 (7.1) | 14 (25.9) | |
| No | 118 (85.5) | 78 (92.9) | 40 (74.1) | |
| Death | | | | 0.711 |
| Yes | 8 (5.8) | 4 (4.8) | 4 (7.5) | |
| No | 129 (94.2) | 80 (95.2) | 49 (92.5) | |

Note:
The bold part of the P value represents P < 0.05.

**Table 5 The logistic regression analysis between biochemical cure and clinicopathological features.**

| | OR | 95% CI | P value |
|---|---|---|---|
| Multifocality | 1.861 | [0.667–5.191] | 0.235 |
| Preoperative calcitonin ≤300 ng/L | 1.969 | [0.626–6.191] | 0.246 |
| Pathologic N classification | | | 0.983 |
| pN1a | 0.954 | [0.061–13.924] | 0.954 |
| pN1b | 0.813 | [0.046–14.464] | 0.888 |
| Resected lymph nodes ≤10 | 1.998 | [0.481–8.302] | 0.341 |
| Metastasized lymph nodes ≤10 | 1.629 | [0.371–7.150] | 0.518 |
| Lymph node ratio ≤0.24 | 2.532 | [0.730–8.787] | 0.143 |
| AJCC clinical stage, III/IV stages | 4.965 | [0.367–67.265] | 0.228 |

Note:
OR, odds ratio; CI, confidence interval.

**Table 6 Univariate and multivariate Cox regression models for predicting disease-free survival.**

| Variables | Univariate analysis | | | Multivariate analysis | | |
|---|---|---|---|---|---|---|
| | HR | 95% CI | P value | HR | 95% CI | P value |
| Age ≤50 years | 0.471 | [0.206–1.077] | 0.074 | | | |
| Gender, male | 0.661 | [0.296–1.478] | 0.313 | | | |
| Tumor size >2 cm | 2.175 | [0.964–4.907] | 0.061 | | | |
| Multifocality | 2.153 | [0.953–4.865] | 0.065 | | | |
| Extrathyroidal extension | 3.146 | [1.345–7.359] | **0.008** | 1.116 | [0.138–9.001] | 0.918 |
| Bilateral | 0.779 | [0.183–3.324] | 0.736 | | | |
| Preoperative calcitonin ≤300 ng/L | 8.120 | [1.898–34.743] | **0.005** | 1.728 | [0.257–11.616] | 0.574 |
| Pathologic T classification, pT3/T4 | 3.531 | [1.463–8.519] | **0.005** | 1.664 | [0.195–14.178] | 0.641 |
| Pathologic N classification, pN0/N1a | 6.075 | [2.267–16.281] | **<0.001** | 1.482 | [0.152–14.483] | 0.735 |
| Resected lymph nodes ≤10 | 7.412 | [2.208–24.874] | **0.001** | 3.242 | [0.356–29.546] | 0.297 |
| Metastasized lymph nodes ≤10 | 3.516 | [1.491–8.290] | **0.004** | 0.469 | [0.153–1.434] | 0.184 |
| Lymph node ratio ≤0.24 | 7.971 | [2.708–23.463] | **<0.001** | 4.818 | [1.270–18.276] | **0.021** |
| AJCC clinical stage, III/IV stages | 16.676 | [2.251–123.546] | **0.006** | 1.128 | [0.071–17.965] | 0.932 |
| Biochemical cure | 4.397 | [1.686–11.468] | **0.002** | 1.486 | [0.512–4.316] | 0.467 |

Notes:
HR, hazard ratio; CI, confidence interval. The bold part of the P value represents P < 0.05.

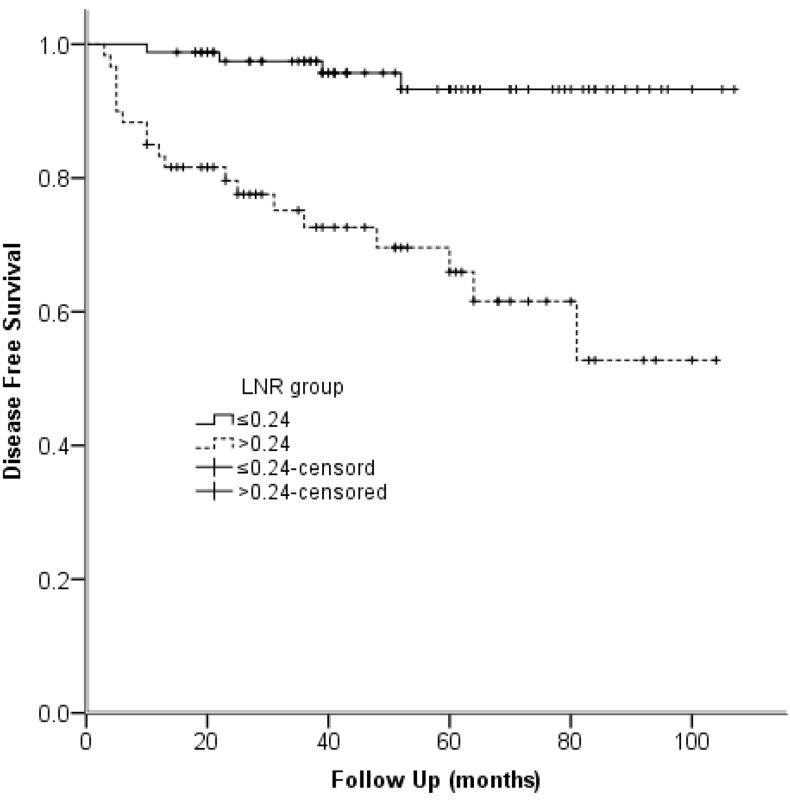

**Figure 2 Kaplan-Meier survival plots present disease-free survival stratified by LNR group.** Note: *P* < 0.001.

**Table 7 Univariate and multivariate Cox regression models for predicting overall survival.**

| Variables | Univariate analysis | | | Multivariate analysis | | |
|---|---|---|---|---|---|---|
| | HR | 95% CI | *P* value | HR | 95% CI | *P* value |
| Age ≤50 years | 0.785 | [0.252–2.439] | 0.675 | | | |
| Gender, male | 0.606 | [0.191–1.919] | 0.394 | | | |
| Tumor size >2 cm | 4.385 | [1.375–13.989] | **0.012** | 2.847 | [0.870–9.312] | 0.084 |
| Multifocality | 1.536 | [0.458–5.146] | 0.487 | | | |
| Extrathyroidal extension | 2.965 | [0.887–9.908] | 0.077 | | | |
| Bilateral | 0.649 | [0.083–5.095] | 0.681 | | | |
| Preoperative calcitonin ≤300 ng/L | 6.943 | [0.881–54.724] | 0.066 | | | |
| Pathologic T classification, pT3/T4 | 2.801 | [0.838–9.361] | 0.094 | | | |
| Pathologic N classification, pN0/N1a | 14.947 | [1.922–116.264] | **0.010** | 4.521 | [0.516–39.627] | 0.173 |
| Resected lymph nodes ≤10 | 64.123 | [0.597–6890.514] | 0.081 | | | |
| Metastasized lymph nodes ≤10 | 3.251 | [0.971–10.884] | 0.056 | | | |
| Lymph node ratio ≤0.24 | 15.994 | [2.063–124.023] | **0.008** | 10.061 | [1.222–82.841] | **0.032** |
| AJCC clinical stage, III/IV stages | 43.503 | [0.332–5694.725] | 0.129 | | | |
| Biochemical cure | 1.870 | [0.467–7.488] | 0.376 | | | |

**Notes:**
HR, hazard ratio; CI, confidence interval. The bold part of the *P* value represents *P* < 0.05.

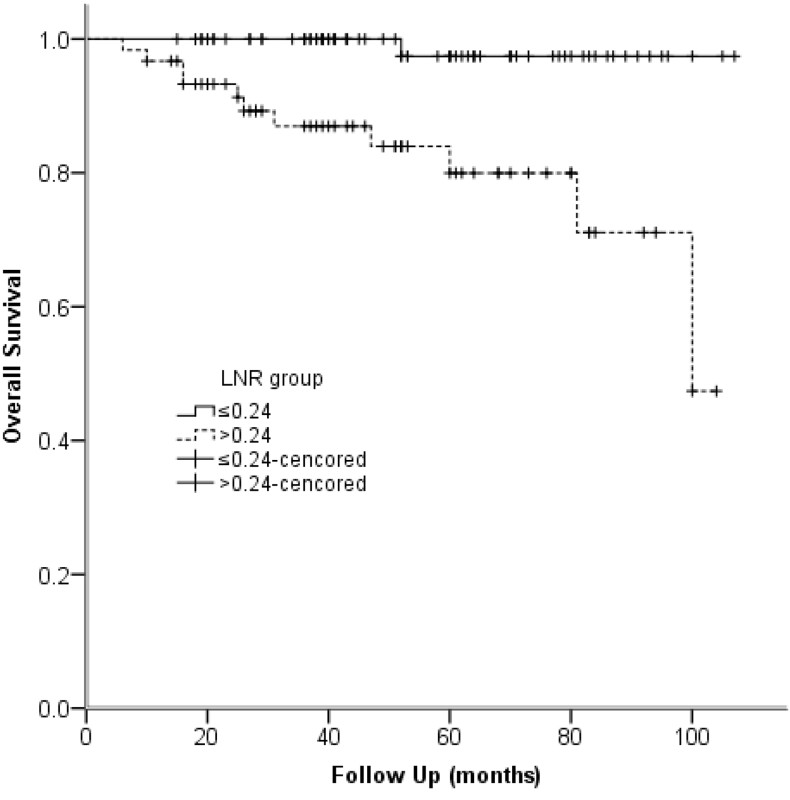

**Figure 3 Kaplan–Meier survival plots present overall survival stratified by LNR group.** Note: $P < 0.001$.

## Prognostic factors influencing disease-free survival and overall survival

We used univariable and multivariable Cox regression models to identify the clinical characteristics affecting structural recurrence. In univariable analyses, gross extrathyroidal extension, preoperative calcitonin levels, pathologic T classification, pathologic N stage, resected lymph nodes, LNM, LNR, AJCC clinical stage, and biochemical cure were significant ($P < 0.05$) factors of DFS. When the multivariable analysis was performed based on the meaningful variables selected from univariate regression, LNR was identified as predictor of DFS (HR = 4.818, 95% CI [1.270–18.276]; $P = 0.021$) (Table 6). The Kaplan-Meier plot of DFS for LNR is provided in Fig. 2.

Univariable Cox regression models reflected that tumor size, pathologic N stage, and LNR were identified as predictors of OS. Furthermore, multivariable analysis manifested that LNR was predictor of OS (HR = 10.061, 95% CI [1.222–82.841]) (Table 7). The Kaplan-Meier plot of OS for LNR is provided in Fig. 3.

## DISCUSSION

Previous studies have indicated that resected lymph nodes number, metastatic lymph nodes number, and ratio of metastatic lymph nodes to the total number of lymph nodes resected tended to be associated with survival outcomes in MTC patients (*Leggett et al.,*

*2008*; *Machens & Dralle, 2013*; *Moses et al., 2021*). Whereas the current AJCC TNM classifications for MTC categorizes lymph node metastases, not by number but location of metastatic nodes. Patients belonging to the same pathologic N stage do not have equal disease burden. Thus, the American Thyroid Association Task Force suggested that lymph node status should be incorporated into the AJCC staging systems for predicting outcomes and planning long-term follow-up of MTC patients (*Wells et al., 2015*).

The present retrospective study aimed to investigate the role of resected lymph nodes, LNM, and LNR for predicting biochemical and structure recurrence in MTC. Multifocality, preoperative calcitonin levels, pathologic N stage, resected lymph nodes, LNM, LNR, and AJCC clinical stage were significant prognostic factors influencing biochemical cure. In addition, we found LNR was an independent prognostic factor of DFS and OS.

The current guidelines for MTC lack a specific lymph node number to guarantee the adequacy of the lymph node dissection and cannot reflect the effects of surgery. Thus, the number of resected nodes, LNM as well as LNR might provide more meaningful prognostic information for MTC patients who undergo surgery. In a previous study that enrolled 2,627 MTC patients, the number of positive nodes was divided into four groups, 0, 1 to 10, 11 to 20, and greater than 20 positive nodes. It manifested patients with 11 to 20 positive central lymph nodes had significantly worse survival than patients with 1 to 10 (*Moses et al., 2021*). Likewise, *Machens & Dralle (2013)* came to the same conclusion. Consequently, we classified both resected and metastatic lymph nodes into two groups, 0 to 10, and greater than ten nodes considering our fewer samples than the researchers above.

In our study, the chi-squared analysis indicated that resected lymph nodes, LNM, and LNR were significant prognostic factors influencing biochemical cure (Table 4). While, logistic regression analysis did not get positive results (Table 5). More samples may be available to get more profound effects. Nevertheless, multiple studies have found that postoperative serum calcitonin is a significant prognostic factor (*Grozinsky-Glasberg et al., 2007*; *Yang et al., 2015*). Therefore, the status of nodes may also be used in combination with postoperative calcitonin levels to predict patients' prognosis (*Yip et al., 2011*).

To some extent, the number of resected and metastatic lymph nodes relies on both surgery and pathologic processing. By contrast, the LNR, which is the number of metastatic lymph nodes divided by the number of resected lymph nodes, maybe a better independent prognostic factor regardless of the personal skill level. We used ROC analysis to define the cut-off value of LNR. Finally, we chose 0.24 to differentiate the high- and low-risk groups for structural recurrence. In univariable studies, pathologic N stage, resected lymph nodes, LNM, and LNR were significant ($P < 0.05$) prognostic factors of DFS (Table 6). Furthermore, multivariable analysis manifested LNR was an independent predictor of DFS (HR = 4.818, 95% CI [1.270–18.276]; $P = 0.021$). Figure 2 demonstrates DFS between high-risk and low-risk series. Moreover, 5-year DFS was 93.2% and 65.9% in different risk groups. *Rozenblat et al. (2020)* and *Jiang et al. (2017)* reached an agreement with our study. By contrast, several previous studies have different LNR cut-off values varied from 0.10 to 0.50 (*Kim et al., 2021*; *Qu et al., 2016*; *Rozenblat et al., 2020*). Therefore,

studies with a more extended follow-up period and a larger population are needed to determine the optimal cut-off value of LNR. What's more, LNR is calculated right after the initial treatment of surgery. And previous studies focusing on other tumors have found that LNR can serve as a reliable prognostic factor (*Mansour et al., 2018*; *Mizrachi et al., 2013*).

Univariable Cox regression models demonstrated that LNR, pathologic N classification, and tumor size were predictors of OS ($P < 0.05$) (Table 7). Additionally, multivariable analysis manifested that LNR was predictor of OS (HR = 10.061, 95% CI [1.222–82.841]). *Jiang et al. (2017)* also stated that LNR was significantly associated with OS. The Kaplan-Meier plot illustrated that OS in LNR high-risk group was 80.0% at 5 years and 97.4% in the low-risk group (Fig. 3).

The present study found that LNR had the strongest association with DFS and OS, which is consistent with the previous studies. Meanwhile, LNR was a predictor of biochemical cure. These findings may help make up a revised staging classification that incorporates the status of nodes.

The limitation of this study is its retrospective design at a single center. Additionally, we did not include all patients with MTC, instead limiting our survey to those sporadic MTC patients. Finally, more patients and more extended follow-up periods are needed.

## CONCLUSION

In conclusion, this study illustrated that LNR was independent prognostic factor of DFS and OS in MTC. In addition, LNR influenced biochemical cure. Further investigations are needed to determine the optimal cut-off value for predicting prognosis.

## ACRONYMS

**MTC**      Medullary thyroid carcinoma
**AJCC**     American Joint Committee on Cancer
**LNM**      Metastatic lymph nodes
**LNR**       Lymph node ratio
**DFS**       Disease-free survival
**OS**        Overall survival

### Funding

This work was supported by grants from the National Natural Science Foundation of China (82172821, 82103386), the Tianjin Municipal Science and Technology Project (19JCYBJC27400, 21JCZDJC00360), the Beijing-Tianjin-Hebei Basic Research Cooperation Project (20JCZXJC00120), The Science & Technology Development Fund of Tianjin Education Commission for Higher Education (2021ZD033), the Tianjin Medical Key Discipline (Specialty) Construction Project (TJYXZDXK-058B), the Tianjin Health Research Project (TJWJ2022XK024), the Tianjin Medical University Cancer Institute & Hospital innovation research fund (2002) and the Tianjin Binhai New Area Health

Commission Project (2022BWKQ027). The funders had no role in study design, data collection and analysis, decision to publish, or preparation of the manuscript.

## Grant Disclosures

The following grant information was disclosed by the authors:

National Natural Science Foundation of China: 82172821, 82103386.

Tianjin Municipal Science and Technology Project: 19JCYBJC27400, 21JCZDJC00360.

Beijing-Tianjin-Hebei Basic Research Cooperation Project: 20JCZXJC00120.

Science & Technology Development Fund of Tianjin Education Commission for Higher Education: 2021ZD033.

Tianjin Medical Key Discipline (Specialty) Construction Project: TJYXZDXK-058B.

Tianjin Health Research Project: TJWJ2022XK024.

Tianjin Medical University Cancer Institute & Hospital Innovation Research Fund: 2002.

Tianjin Binhai New Area Health Commission Project: 2022BWKQ027.

## Competing Interests

The authors declare that they have no competing interests.

## Author Contributions

- Weijing Hao conceived and designed the experiments, performed the experiments, analyzed the data, prepared figures and/or tables, authored or reviewed drafts of the article, and approved the final draft.
- Jingzhu Zhao conceived and designed the experiments, performed the experiments, analyzed the data, prepared figures and/or tables, authored or reviewed drafts of the article, and approved the final draft.
- Fengli Guo performed the experiments, analyzed the data, prepared figures and/or tables, and approved the final draft.
- Pengfei Gu performed the experiments, analyzed the data, prepared figures and/or tables, and approved the final draft.
- Jinming Zhang performed the experiments, analyzed the data, prepared figures and/or tables, and approved the final draft.
- Dongmei Huang performed the experiments, analyzed the data, prepared figures and/or tables, and approved the final draft.
- Xianhui Ruan performed the experiments, prepared figures and/or tables, and approved the final draft.
- Yu Zeng performed the experiments, prepared figures and/or tables, and approved the final draft.
- Xiangqian Zheng conceived and designed the experiments, analyzed the data, prepared figures and/or tables, authored or reviewed drafts of the article, and approved the final draft.
- Ming Gao conceived and designed the experiments, analyzed the data, prepared figures and/or tables, authored or reviewed drafts of the article, and approved the final draft.

## Human Ethics

The following information was supplied relating to ethical approvals (*i.e.*, approving body and any reference numbers):

The present study was approved by the Institutional Review Board of Tianjin Medical University Cancer Institute and Hospital (bc2022191).

## Data Availability

The raw measurements are available in the Supplemental File.

## Supplemental Information

Supplemental information for this article can be found online at http://dx.doi.org/10.7717/peerj.15025#supplemental-information.

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
