# Peer review of "Value of lymph node ratio as a prognostic factor of recurrence in medullary thyroid cancer"

_PeerJ, doi:10.7717/peerj.15025_

## Round 0.1 · original submission · Major Revisions

Dear authors,

As you will see, the 2 reviewers found interesting points in your paper but recommend major revisions.

Please address ALL the issues raised by the reviewers, in the point-by-point rebuttal letter

In addition please check the figure numbers in the manuscript and in the review document.

Reviewer 1 ·

Basic reporting

The manuscript aims at evaluating the relationship between lymph node status (the number of resected lymph nodes, the number of metastatic lymph nodes, and lymph node ratio) and biochemical recurrence, disease-free survival, as well as overall survival. The study investigated 160 medullary thyroid carcinoma patients. And illustrated that LNR was an independent prognostic factor of disease-free survival in medullary thyroid carcinoma. In addition, LNR influenced biochemical cure and overall survival.
The language of this manuscript is clear and professional. The introduction offered a good amount of information about the field. The literature was well-referenced and relevant. However, it would be better if the author could give more background information (eg. the mechanism and importance) about lymph node metastasis and medullary thyroid carcinoma for the readers. The manuscript was properly structured.
The tables and figures offered sufficient data to support the conclusions. However, the format of the tables and figures can be improved.
In line 111, the authors mentioned “table 2-3”. It was not clear whether the authors would like to mention a part of table 2 (There are no tables labeled 2-3) or “table 2 and 3”.
In the manuscript, only Fig.1 to Fig. 3 were mentioned. But in the uploaded files, there are 5 figures. Moreover, the title and the content of the figures are not matched well. It looks like Fig. 1 in the text, is Fig. 1 and 2 in the appendix. Fig. 2 in the text is Fig. 3 in the appendix. Fig. 3 in the text is Fig. 4 and 5 in the appendix. In addition, it would be better if the author can offer more information in the figure captions.
Fig. 1 and 2 (in the appendix), the x-axises are missing units. Also, it would be more clear to the reader if the authors can merge these two graphs.
Although the acronyms are provided by the authors, it would still be helpful to the readers if the authors could mention the full name before using the acronyms. For example, LNR was mentioned in line 61. But prior to that, there is no full name mentioned. Actually, the author can easily specify that in line 59 when mentioning “lymph node ratio”. Similar problems with LN in line 98 and 99.
In addition, when the authors mentioned the full names of the acronyms, sometimes the author still uses the full name (Eg. line 143, “disease-free survival”; line 144, “overall survival”)

Experimental design

The research questions were defined well, relevant and meaningful. The methods were described in sufficient detail.

Validity of the findings

The authors pointed out the lymph node ratio is an independent prognostic factor of disease-free survival. Such a finding suggests that more research is needed to discover the mechanism behind it.

Reviewer 2 ·

Basic reporting

1. In section 3.4, “In univariable analyses, gross extrathyroidal extension, preoperative calcitonin levels, pathologic T classification, pathologic N stage, resected lymph nodes, LNM, LNR, AJCC clinical stage, and biochemical cure were significant (P<0.05) factors.” Was the response DFS or OS?

Experimental design

1. Why patients with a family history of MTC were excluded from the study?

Validity of the findings

1. When specifying the predictors, it seems that the authors applied univariate regression based on variables selected from univariate regression? Please clarify.
2. Did the authors use 0.05 as the cutoff for univariate analysis? Could it be too small?
3. Why didn’t the authors apply multivariate analysis for OS?

---

## Round 0.2 · accepted · Accept

Thank you for the modifications done in your manuscript.
The 2 Reviewers are now positive about your work.

However there is missing the p value in the Kaplan Meyer figures 2 and 3. Please add that information to the graph or in the legend of the figures.

Thank you for submitting your work to PeerJ

Reviewer 1 ·

Basic reporting

The authors have addressed the points well and corrected them well.

Experimental design

No comment.

Validity of the findings

No comment.

Reviewer 2 ·

Basic reporting

No comment.

Experimental design

No comment.

Validity of the findings

No comment.